# Antimicrobial Activity of Silver and Gold Nanoparticles Prepared by Photoreduction Process with Leaves and Fruit Extracts of *Plinia cauliflora* and *Punica granatum*

**DOI:** 10.3390/molecules27206860

**Published:** 2022-10-13

**Authors:** Marcia Regina Franzolin, Daniella dos Santos Courrol, Flavia Rodrigues de Oliveira Silva, Lilia Coronato Courrol

**Affiliations:** 1Laboratório de Bacteriologia, Instituto Butantan, São Paulo 05503-900, Brazil; 2Centro de Ciências e Tecnologia dos Materiais, Instituto de Pesquisas Energéticas e Nucleares, IPEN/CNEN-SP, São Paulo 05508-900, Brazil; 3Instituto de Ciências Ambientais, Químicas e Farmacêuticas, Departamento de Física, Universidade Federal de São Paulo, Diadema, São Paulo 04023-062, Brazil

**Keywords:** *Plinia cauliflora*, *Punica granatum*, antibacterial activity, gold nanoparticles, silver nanoparticles, photoreduction

## Abstract

The increased number of resistant microbes generates a search for new antibiotic methods. Metallic nanoparticles have emerged as a new platform against several microorganisms. The nanoparticles can damage the bacteria membrane and DNA by oxidative stress. The photoreduction process is a clean and low-cost method for obtaining silver and gold nanoparticles. This work describes two original insights: (1) the use of extracts of leaves and fruits from a Brazilian plant *Plinia cauliflora,* compared with a well know plant *Punica granatum,* and (2) the use of phytochemicals as stabilizing agents in the photoreduction process. The prepared nanoparticles were characterized by UV-vis, FTIR, transmission electron microscopy, and Zeta potential. The antimicrobial activity of nanoparticles was obtained with Gram-negative and Gram-positive bacteria, particularly the pathogens *Staphylococcus aureus* ATCC 25923; *Bacillus subtilis* ATCC 6633; clinical isolates of methicillin-resistant *Staphylococcus aureus* (MRSA) and *Enterococcus faecalis*; *Escherichia coli* ATCC 25922; *Escherichia coli* O44:H18 EAEC042 (clinical isolate); *Klebsiella pneumoniae* ATCC 700603, *Salmonella Thiphymurium* ATCC 10231; *Pseudomonas aeruginosa* ATCC 27853; and *Candida albicans* ATCC 10231. Excellent synthesis results were obtained. The AgNPs exhibited antimicrobial activities against Gram-negative and Gram-positive bacteria and yeast (80–100%), better than AuNPs (0–87.92%), and may have the potential to be used as antimicrobial agents.

## 1. Introduction

The world lives in the era of pandemic diseases and needs alternative prophylactic or therapeutic tools against microbes [1]. Antibiotic resistance is a cause of concern that continues to challenge the healthcare field in several parts of the world, affecting both developing and developed countries [2,3]. Because of the emergence and spread of multidrug-resistant pathogens, it is necessary to search for new antimicrobial substances to treat infectious diseases.

The antibacterial and antiviral actions of metal nanoparticles, notably silver nanoparticles (AgNPs), are well-known [4,5,6,7]. The application of nanoparticles provides a potential strategy to control infections caused by multidrug-resistant organisms [8]. Gram-negative bacteria, such as *Escherichia coli*, *Pseudomonas aeruginosa*, and *Klebsiella pneumoniae*, and Gram-positive bacteria, such as *Staphylococcus aureus* and *S. epidermidis*, are principally responsible for hospital-acquired infections.

Nanoparticles have an antimicrobial activity that can overcome common resistant mechanisms, such as the modification of a drug target, enzyme inactivation, cell permeability, and active efflux to escape from the antibacterial activity of antimicrobial agents [9,10,11]. The AgNPs disrupt the integrity of the multidrug-resistant bacteria wall and membrane, promote lipid peroxidation and oxidative damage of DNA and proteins, induce damage and aggregation of the DNA, disrupting its transcription and translation, interfering with the process of cell signal transduction, which finally kills the cells [12,13,14,15].

The bottom-up approach to synthesizing metal nanoparticles involves reducing metal ions from their ionic salts by using chemical reducing agents in the presence of a stabilizing agent under favorable reaction parameters (pH, temperature, etc.). An extensive list of some reducing agents is available for this process. Synthesis of single or multi-component noble metal NPs formed using various chemical and green synthesis methods (biomolecules including DNA, protein, enzyme, and plant extracts) with their biological properties have been studied [16,17,18,19].

It is possible to enumerate several essential applications of silver nanoparticles (AgNPs): antibacterial, antifungal, antiviral, and antitumoral agents, diagnostic applications when used in biosensors, conductive and optical applications, etc. [7,20,21,22].

Gold nanoparticles (AuNPs) are easy to synthesize and have attracted extensive attention since they can be applied to a wide range of medical applications, including drug and gene delivery, photothermal therapy, photodynamic therapy, diagnosis, X-ray imaging, computed tomography, and other biological activities [23,24,25].

Plants extracts have been employed to synthesize nanoparticles making the process inexpensive and eco-friendly [26,27,28,29]. In 2003 plant-based synthesis of silver nanoparticles was published using *Medicago sativa* [30]. After this publication, several other studies were made using leaves or fruit extracts to obtain silver nanoparticles [31,32,33,34].

The plant extract-based gold nanoparticle (AuNP) synthesis was also demonstrated, and there are hundreds of plants employed for their extracts to synthesize AuNPs [35,36,37,38,39,40]. Plant-based AuNPs have also been reported as antimicrobial agents [41,42,43,44,45,46,47].

The phytochemical agents present in plant extracts can reduce silver or gold to produce nanoparticles [48,49,50,51,52]. The advantages of plant extract-based nanoparticle synthesis are the low-cost, safety, straightforward protocol, the nanoparticles have high stability, generate non-toxic by-products, and allow large-scale synthesis. Some disadvantages can be listed: the process is not controlled, and the particles are not monodispersed.

The photoreduction process employs light to promote the photochemical reaction and reduce metal ions to zero-valent metal, using the photochemically generated intermediates, such as excited molecules and radicals [53]. The advantages of this method are the absence of reducing agents, high spatial resolution, controllable generation of nanoparticles, and great versatility. This work employed the photoreduction process to synthesize silver and gold nanoparticles, and plant extracts were used as stabilizing agents.

The Jabuticabeira (*Plinia cauliflora* (Mart.) Kausel) is a tree of the family *Myrtaceae*, which includes about 5900 species and is found in the Atlantic rainforest, Pantanal, Cerrado, and Caatinga biomes in Brazil [54]. It is a fruit tree with a height between 10 to 15 m, with a smooth trunk of 30–40 cm in diameter. The leaves are approximately seven centimeters long. It blooms in spring and summer. Fruits grow in clusters on the trunk and branches. Fruits have a black peel and white pulp adhered to the only seed, are consumed mainly in natura, or in the form of jam, juice, liquor, brandy, wine, and vinegar. The dark bark contains pectin and peonidin in addition to a pigment, anthocyanin, responsible for the blue-purplish coloration of jabuticaba [55,56].

Fruits and leaves of *P. cauliflora* present antioxidant, anti-inflammatory, antimicrobial, and antiproliferative activities [55,56,57,58,59]. Jabuticaba is used for the treatment of diarrhea, asthma, chronic inflammation of the tonsils, etc. [59,60].

*Punica granatum L*., is a tree of *Lathraceae* family, native to Asia. Its fruit is commonly known as the pomegranate. The leaves, bark, and fruit have medicinal applications in traditional medicine in many countries for treating diarrhea, conjunctivitis, helminthiasis, and hemorrhages [61]. This plant contains phytochemicals, such as terpenoids, alkaloids, sterols, polyphenols, sugars, fatty acids, aromatic compounds, amino acids, tocopherols, etc.

Previous studies reported *Punica granatum* pomegranate extract as a good reducing agent to synthesize silver nanoparticles [62,63,64] and gold nanoparticles [61,65,66]. Strong antibacterial action was observed for pomegranate AgNPs on either Gram-positive or Gram-negative bacteria [67,68,69,70]. Furthermore, the cytotoxic effects on cancer cell lines were tested [67,70].

In this study, the aqueous extracts of leaves and fruit from *P. cauliflora* and *P. granatum* were used to synthesize silver and gold nanoparticles by photoreduction. The nanoparticles were characterized by UV-Vis, TEM, Zeta-potential, and FTIR. Moreover, the antimicrobial activity of synthesized nanoparticles was evaluated through an in vitro investigation.

## 2. Materials and Methods

### 2.1. Materials and Reagents

*Plinia cauliflora* and *Punica granatum* leaves were collected from a spontaneous germination tree in São Paulo, SP, Brazil. Silver nitrate and tetrachloroauric(III) acid were acquired from Sigma-Aldrich. All solutions were prepared with double-distilled water.

### 2.2. Preparation of Nanoparticles

Leaves and fruits of *Plinia cauliflora* (Pc) and *Punica Granatum* (Pg) were repeatedly washed with double distilled water, chopped, weighed (1.040 ± 0.025 g), and boiled in 40 mL of doubly distilled water until they reached the temperature of 80 ± 2 °C. The pH of obtained extracts was ~6.5. After 1 min, the solutions were filtered, and ~0.6 mM of AgNO_3_ was added to hot extracts to prepare silver nanoparticle solutions (PcAgNPs and PgAgNPs). An aliquot of 10 mL of solutions was immediately exposed to a 300-Watt Cermax Xenon lamp for 1 min (3.6 W/cm^2^) to improve nanoparticle properties. The ventilation system of Xe lamp promoted a little agitation in solution during synthesis. Immediately after the photoreduction process, pH was adjusted to neutral with NaOH (1M). For PcAgNPs prepared with extracts of leaf extracts by photoreduction, a variation in AgNO_3_ concentration (0–0.6 mM) was tested. The entire synthesis process from the phytochemical extraction until pH adjustment can be performed in around 10 min. To prepare gold nanoparticles (PcAuNPs and PgAuNPs), leaves and fruits (2.200 ± 0.025 g) were boiled (until 80 ± 2 °C) in 40 mL of doubly distilled water. The solutions were exposed to a 300 Watt Cermax Xenon lamp for 1 min. After the photoreduction process, pH was adjusted to neutral. To compare PcAuNPs and PgAuNPs prepared with leaves and fruit extracts was used 25 mM of HAuCl_4_. For PcAuNPs prepared with extracts of leaf extracts, a variation in HAuCl_4_ concentration (0–25 mM) was tested. The synthesis mechanism is illustrated in Figure 1.

### 2.3. Characterization of Nanoparticles

Spectrophotometry analyses in the UV-vis region were performed with the UV-vis Shimatzu MultiSpec 1501 spectrophotometer. The measurements were carried out in an optical path quartz cuvette (10 mm) in the 200 and 800 nm range. To perform UV-Vis analysis, AgNPs were diluted 20 times in double-distilled water, and the AuNPs were measured as prepared. Fourier-transform infrared spectroscopy (FTIR) was obtained with a Shimatzu IRPrestige. A volume of 200 µL of the extracts was deposited in microscope slices and left at room temperature until dry. The procedure was repeated three times. KBr pellets were prepared with grated powder. The stability of the colloidal suspension was analyzed by Zeta potential measurements using the Zetasizer Nano ZS Malvern. Three assays were made for each sample. The JEM 2100 JEOL microscope obtained transmission electron microscopy images.

### 2.4. Determination of Antibacterial Efficacy of Plant Extracts and NPs

The microorganisms employed in this assay were: *Staphylococcus aureus* American Type Culture Collection (ATCC) 25923; *Bacillus subtilis* ATCC 6633; clinical isolates of methicillin-resistant *Enterococcus faecalis* and *Staphylococcus aureus* (MRSA); *Escherichia coli* ATCC 25922; *Escherichia coli* O44:H18 EAEC042 [71] (clinical isolate); *Klebsiella pneumoniae* ATCC 700603, *Salmonella Thiphymurium* ATCC 14028; *Pseudomonas aeruginosa* ATCC 27853; and *Candida albicans* ATCC 10231. The resistance profile (clinical strains): MRSA-amikacin, clindamycin, gentamicin, oxacillin, penicillin, and sulfonamide; *E. faecalis*-amikacin, gentamicin, clindamycin, oxacillin, penicillin, and erythromycin; *E. coli* O44:H18 EAEC042-chloramphenicol, tetracycline, and streptomycin. The antimicrobial activity of the plant extracts, AgNPs, and AuNPs was tested in triplicate in 96-well microtiter plates, according to the CLSI guidelines (CLSI, 2015). A volume of 50 µL of Mueller-Hinton (MH) broth (bacteria) or Sabouraud (Sab) broth (*C. albicans*), with bacterial or fungal inoculum with 10^6^ CFU/mL, was added to 50 µL of plant extracts, and the nanoparticle solutions which were diluted in MH or Sab broth, resulting in a volume of 100 µL with 10^4^ CFU/well. Next 20 h of incubation at 37 °C, microbial growth was measured by observing the optical density (OD) changes at 595 nm in an enzyme-linked immunosorbent assay (ELISA) reader (Multiskan^®^EX-Thermo Fisher Scientific, EUA). The results were stated as inhibition percentage of OD against a control (microorganisms in the absence of AgNPs). The rate of microbial inhibition was calculated by the following formula:(1)$$\%\mathrm{Cell}\;\mathrm{inhibition}=\frac{\left[\mathrm{Control}\;\mathrm{at}\;OD_{595\mathrm{nm}}-\;\mathrm{Test}\;\mathrm{at}\;OD_{595\mathrm{nm}}\right]\;}{\mathrm{Control}\;\mathrm{at}\;OD_{595\mathrm{nm}}}\times 100$$

### 2.5. Statistical Analysis

Results were significant when *p* < 0.05 by the Student’s *t*-test.

## 3. Results

### 3.1. Plants Extracts

Figure 2 shows the UV-Vis spectra obtained for Pc and Pg leaf and fruit extracts. The extracts present bands in the UV region attributed to plant pigments such as polyphenols, including flavonoids, tannic acid, and ellagitannin. It is possible to note that fruit extracts present a higher concentration of phenolic compounds such as ellagic acid and anthocyanin than leaf extracts, and Pg extracts have more intense bands than Pc.

The irradiation on the phytoextract by 1 min does not promote a significant change in UV-Vis spectra.

### 3.2. NPs Characterization

Silver nanoparticles exhibit maximum absorption in the 400–500 nm range due to surface plasmon resonance (SPR). The SPR band intensity indicates NP concentration, and position and width at half height indicate particle size and homogeneity. Figure 3a shows the UV-Vis spectra for AgNPs prepared by photoreduction with Pc leaf extract with increased AgNO_3_ concentration. An increase in SPR band intensity and a bathochromic shift is observed with the increase in AgNO_3_ concentration. Figure 3b,c show the UV-Vis spectra of PcAgNPs prepared with leaves and fruit extracts before and after the photoreduction process. Bands observed in spectra suggest the formation of the PcAgNPs after adding AgNO_3_ to the extracts. After the photoreduction process and pH adjustment to 7.0, the intensity of SPR bands increased, and the bands became narrow (416 nm and 398 nm leaf and fruit extract, respectively), indicating more homogeneous particles, especially PcAgNPs prepared with fruit extract (TEM image showing particles with ~8–20 nm).

Figure 4a,b show the results obtained with the synthesis of PgAgNPs prepared with leaves and fruit extract, respectively. For nanoparticles prepared with *P. granatum*, the observed SPR bands were large, with double peaks with maxima around 400/530 nm and 397/540 nm for NPs prepared with extracts of leaves and fruit, respectively. Broad peaks in the spectra indicate that solutions are inhomogeneous due to agglomerates. After the photoreduction process and pH control, the SPR band shifted to 412/414 nm for NPs prepared with extracts of leaves and fruit, respectively.

Figure 5a shows the UV-Vis spectra obtained for PcAuNPs (leaf extract) prepared by photoreduction and pH control with an increasing gold concentration. With the increase in gold concentration, the SPR band intensity increased, and the band became narrower with a slight blue shift from ~537 to 530 nm, indicating the presence of more homogenous solutions. The TEM image of PcAuNP indicates the existence of spherical nanoparticles with diameters between 2 and 12 nm, approximately. Figure 5b compares PcAuNPs and PgAuNPs prepared with leaf extracts before and after the photoreduction process. It is observed a shift od SPR band from 530 nm to 526 nm for PcAuNPs and 560 to 534 nm for PgAuNPs, indicating that although the intensity remains practically unchanged, the nanoparticles had a reduction in their sizes and became more homogeneous, mainly PgAuNPs. The same occurred for nanoparticles prepared with fruits extract as observed in Figure 5c. A shift in the SPR band from 530 nm for 526 nm was observed for PgAuNPs and for 557 nm to 547 was observed for PgAuNPs. The increase in the UV-Vis absorption indicate probably the presence of higher concentration of AuCl_4_^−^ species in the solution.

The results obtained for Zeta potential measurements for the PcNPs and PgNPs are shown in Table 1. It is possible to observe that PcAgNPs prepared with leaf extracts before the photoreduction (PR) process presented Zeta potential values of −10.8 mV and after the photoreduction process −15.5 mV, indicating an increase in the stability of the particles. The same was observed for PcAgNPs prepared with fruits extract: −17.1 mV before PR and −26.7 mV after PR. PcAg (−26.7 mV) was more stable than PgAg fruit extract (−22.0 mV), but PgAg prepared with leaf extract was more stable than PcAg (−20.1 and −15.5 mV, respectively). PgAuNPs were more stable than PcAuNPs by the two types of phytoextract. Figure 6 presents the Fourier transform infrared spectroscopy (FTIR) obtained for the phytoextract [72,73] and PcAgNPs and PcAuNPs. The spectra present strong and broad peaks around 3000 to 3600 cm^−1^ corresponding to the -NH_2_ (amide I) and/or -OH of phenolic compounds. The sharp doublet peaks at 2920 cm^−1^ and 2845 cm^−1^ (fruit extract) correspond to the symmetric and asymmetric vibrational mode of -CH stretching. The spectral region between 1600 and 1700 cm^−1^ is related to aromatic C=O stretching vibration of carbonyl. The absorption peak around 1733 cm^−1^ due to the C=O stretching vibration absorption of flavonoids and amides is observed in PcAg prepared leaf extract. After the photoreduction process, there is a reduction in this band intensity, indicating that C=O bond was oxidized [74]. The sharp peak at 1384 cm^−1^ is due to C-H stretching vibrations of aromatic and aliphatic amines. The peak around 1060 cm^−1^ indicates C-O stretching vibrations correspond to the presence of alcohols, carboxylic acids, ethers, and esters. In this way, the IR spectra of prepared NPs thus confirmed that the carbonyl group of polyphenols can bind metal, indicating that the biological molecules could perform both functions of formation and stabilization of silver nanoparticles in the aqueous medium. Comparing the PcNPs measured spectra with standard anthocyanins [75], it can be concluded that the samples contain anthocyanin compounds. [72,75]. FTIR spectra for PcAgNPs were already studied by other authors and present similarities with the spectra shown in Figure 6 [64].

### 3.3. Antibacterial Efficacy Plant Extracts and PcNPs and PgNPs

The antimicrobial activity of extract obtained from Pc leaf and fruit extracts and AgNPs and AuNPs, prepared with leaf and fruit extracts of Pc and Pg, was investigated for Gram-positive bacteria (staphylococcus, enterococcus, and bacillus), Gram-negative bacteria (enterobacteria and non-fermenters) and yeast. Figure 7a shows growth inhibition results obtained with Pc leaf extracts PcAgNPs, PgAgNPs and PcAuNPs prepared with leaf extract. It can be noted that AgNPs promote higher inhibition than AuNPs, and PcAgNPs presented better results than those obtained for PgAgNPs. For PcAgNPs, inhibitions were between 95–100%. For PgAgNPs, inhibitions ranged between ~68% (*E. faecalis*) and 99.5%. The antimicrobial activity of PcAuNPs ranged from 0–84.75%, presenting a high variability of percentual growth inhibition, with a mean of 41.43% ± 23.30%. The growth inhibition of PcAuNPs was comparable to the leaf extract inhibition. Figure 7b shows growth inhibition results obtained with Pc fruits extracts, PcAgNPs, PgAgNPs, PcAuNPs, and PgAuNPs prepared with fruit extract. It can be noted that AgNPs promote higher inhibition than AuNPs, and PcAgNPs presented better results than those obtained for PgAgNPs. For PcAgNPs, inhibitions were higher than 95%. For PgAgNPs, inhibition was higher than 86%. The growth inhibition of PcAuNPs was comparable to those obtained with PgAuNPs and for the Pc leaf extract. High inhibition was observed of Gram-positive, and Gram-negative bacteria and yeast after treatment with PcAgNps (94.97% to 100%, with a mean of 98.70% ± 1.47), in comparison with the antimicrobial inhibition of leaf and fruit extract and PcAuNPs (*p* < 0.05). All treatments tested showed high inhibition percentual (72.65–95.88%) of *B. subtilis*, especially PcAgNPs.

## 4. Discussion

The leaves of *P. cauliflora* present tannins and flavonoids such as ellagic acid, quercetin, and myricetin [76,77]. *P. cauliflora* fruit contains tannins, carotenoids, flavonoids such as anthocyanins and phenolics acids [55,56,59]. Recently, twenty-three compounds of *P. granatum* were identified by LC–MS–MS analysis of ethyl acetate fraction of pomegranate leaf extract, 11 phenolic acids and their derivatives, eight tannins; 3 anthocyanins, and one flavonoid derivative were detected [64]. Different phenolic compounds is found in pomegranate fruits as gallic, chlorogenic, caffeic, ferulic and ellagic acids, catechin, epicatechin, phloridzin, quercetin and rutin [78]. The UV-vis spectra obtained for the extracts of leaves and fruits presented in Figure 2 showed that fruit extract presented an increased concentration of polyphenols, such as ellagic acids (EA) and anthocyanins, then leaf extract and Pg extracts presented increased signal intensities in the UV region. When silver nitrate solution was added to phytoextracts solutions, a color change was observed instantaneously, indicating that the silver ions (Ag+) were converted to a zero-valent (Ag°), and phytochemicals acted as reducing agents. Then, the nucleation process is initiated, followed by the immediate growth phase, forming large nanoparticles. Figure 3 and Figure 4 show that the characteristic SPR bands are wide, indicating the presence of agglomerates. Several factors can affect the synthesis, for example, pH, temperature, the concentration of plant extract and silver nitrate, and others. The photoreduction process can improve nanoparticle size, shape, stability, and dispersivity, allowing their use in various other applications. When NPs solutions (pH ~ 2.5) were submitted to the photoreduction process immediately after mixing extracts to AgNO_3_ or HAuCl_4_, ketones in the polyphenol structure were excited to the singlet excited state (Figure 1). The singlet excited state decayed to the triplet excited state via the intersystem crossing. The excited triplets and subsequent hydrogen abstraction from the corresponding alcohol resulted in ketyl radicals which are powerful reducing agents [53]. These radicals reduced the remaining metal ions in the solution, increasing the number of nanoparticles. The photoreduction process also induced steric repulsion within nanoparticles, which prevented agglomeration, giving rise to a mutual stabilization system. The solution pH, in this case, decreased to <2. Setting pH to 7.0, the AgNPs surfaces reached a higher degree of deprotonation; thus, the negative surface charge increased, increasing the Zeta potential value. Specifically, in the case of AuNPs synthesis, a potential reaction involved the formation of flavonoids-Au^3+^ complexes, hydrogen (H^+^), and chloride (Cl^−^) ions [79]. With UV irradiation, charge transfer from flavonoids to Au^3+^ as ligand-to-metal was induced, and Au^+^Cl^2−^ species and quinone derivatives were produced. Finally, stable zero valent AuNPs were produced ($3{\mathrm{Au}}^{+}{\mathrm{Cl}}_{2}^{-}\to {\mathrm{Au}}^{0}+{\mathrm{Au}}^{3+}{\mathrm{Cl}}_{4}^{-}+2{\mathrm{Cl}}^{-}$). Among the flavonoids, the anthocyanins exhibit great ability to form strong anthocyanins-Au^3+^ complexes acting as an efficient photo-reductant compared to other reductants [79]. In the case of Pc extract, for example, was observed an increased concentration of anthocyanins (Figure 2) than for Pg extracts which can better explain results obtained for PcAuNps compared to PgAuNPs, mainly for fruit extracts. Figure 3, Figure 4 and Figure 5 presented the results obtained for nanoparticles produced by the photoreduction process of *Plinia cauliflora* and *Punica granatum* extracts showing excellent optical properties. The bands became narrower with the photoreduction process enhanced solution homogeneity. The Zeta potential analysis (Table 1) indicated that nanoparticles produced by the photoreduction process have a more considerable number of charges around the particle, leading to greater repulsion and, consequently, lower aggregation. The antibacterial activity of plant extracts and nanoparticles was investigated against gram-positive bacteria (staphylococcus, enterococcus, and bacillus), gram-negative bacteria (enterobacteria and non-fermenters), and yeast. The results are presented in Figure 7. Several studies demonstrated the antimicrobial activity of *P. cauliflora* and *P. granatum* extracts, possibly related to flavonoids and hydrolyzable tannins [77,80,81,82,83]. These compounds disturb cell wall integrity and membrane permeability, inhibiting protein expression and affecting microbial metabolism [84]. Water extracts of *Punica granatum* presented the antimicrobial activity of *S. aureus, E. coli, P. vulgaris, B. subtilis,* and *S. typhi* [85]. In this study, *P. cauliflora* leaf extract promoted higher growth inhibition than fruit extract for *S. aureus*, *E. faecalis*, *E. coli,* and *C. albicans* (20–70%) related to the presence of flavonoids, alkaloids, saponins, and tannins. Growth inhibition promoted by fruit extracts, rich in anthocyanins and ellagitannins, considered the main responsible compounds for the excellent bioactive properties, occurred in a range of 40–80% for *B. subtilis, E. coli* O44:H18 EAEC042, *K. pneumoniae, S. thiphymurium, and P. aeruginosa*. Souza-Moreira et al. [86] observed a weak inhibitory activity of whole fruit extract for *E. faecalis*, *E. coli*, and *Salmonella,* which was attributed to the lower contents of phenolic compounds in the entire fruit in comparison to its peel. High inhibition percentual of Gram-positive and Gram-negative bacteria and yeast was observed after treatment with PcAgNps (94.97% to 100%, with a mean of 98.70% ± 1.47). PgAgNPs exhibited antibacterial activities against Gram-negative and Gram-positive, as well as antifungal activity in the range of 80% to 99.5%. Silver nanoparticles have several mechanisms of antibacterial action. One of these is the capacity to penetrate the cell wall of the bacteria, leading to an increase in cell permeability followed by cell death [13,87,88]. The results are shown in Figure 7 for AuNPs. By the obtained results, most microbial species tested showed higher inhibition percentages when exposed to fruit extract than leaf extract (*p* < 0.05). In contrast, the opposite occurred with *E. coli* e *C. albicans*. Both plant extracts showed the highest activity against Gram-positive bacteria. The antimicrobial activity of PcAuNPs ranged from 0–84.75%, presenting a high variability of percentual growth inhibition, with a mean of 41.43% ± 23.30%. Despite the low growth inhibition promoted by the produced AuNPs, these nanoparticles have the potential to be applied in medicine, such as drug delivery in cancer treatment. All treatments tested showed high inhibition percentages (72.65–95.88%) of *B. subtilis*, especially PcAgNPs. The results show that silver nanoparticles synthesized with extracts of *Plinia cauliflora and P. granatum* leaves and fruits inhibited the microbial growth of multidrug-resistant and susceptible bacteria strains and yeast. These nanoparticles may help prevent and treat several infections in this form.

## 5. Conclusions

In conclusion, in this study, we presented a simple, fast, and low-cost process for controlled synthesis of nanoparticles from leaf and fruit extracts of *Plinia cauliflora* and *Punica granatum* at room temperature. The photoinduced process and pH adjustment improved particle quality and stability. PcAgNPs and PgAgNPs showed high antimicrobial activity against Gram-negative and Gram-positive bacteria and yeast, whereas PcAuNP and PgAuNPs presented a lower antimicrobial activity. The antibacterial activity of the synthesized NPs further confirms the antibiotic efficiency for developing new antibacterial agents for treatment against Gram-negative and Gram-positive pathogens.

## Figures and Tables

**Figure 1 molecules-27-06860-f001:**
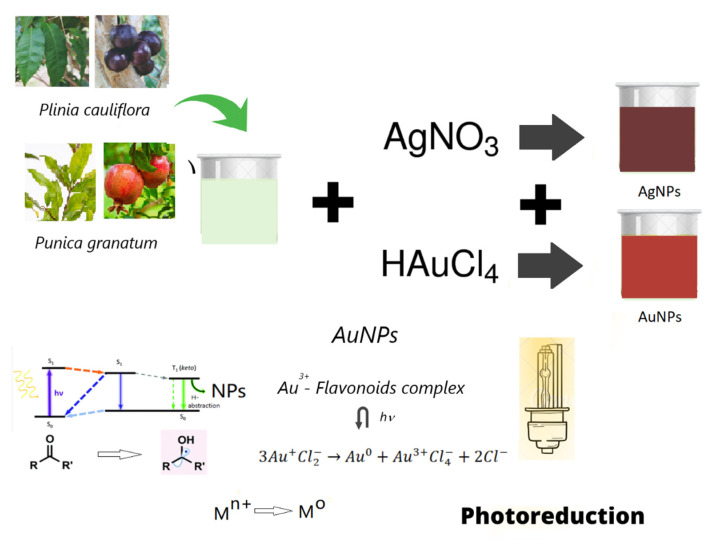
Experimental procedure. Preparation of the Pc and Pg extracts of leaves and fruits, adding AgNO_3_ or HAuCl_4_ in solutions. Intermolecular photoreduction of polyphenol ketones and ketyl Radical Generation. These radicals reduce metal ions to generate metal nanoparticles.

**Figure 2 molecules-27-06860-f002:**
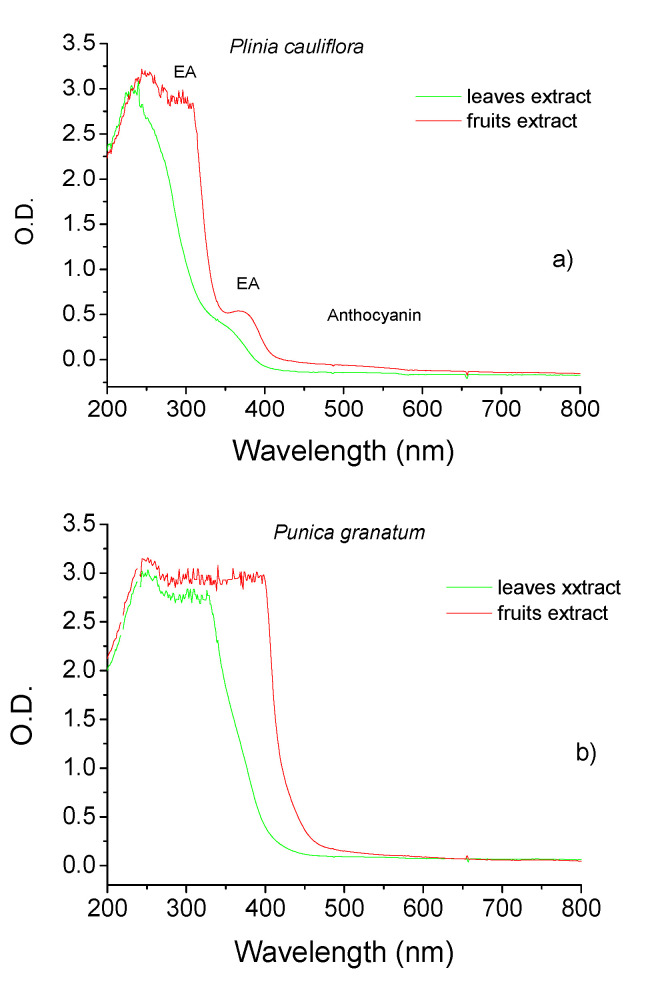
UV-vis spectra of (**a**) Leaves and fruit extracts of *Plinia cauliflora* and (**b**) Leaves and fruit extracts of *Punica granatum*.

**Figure 3 molecules-27-06860-f003:**
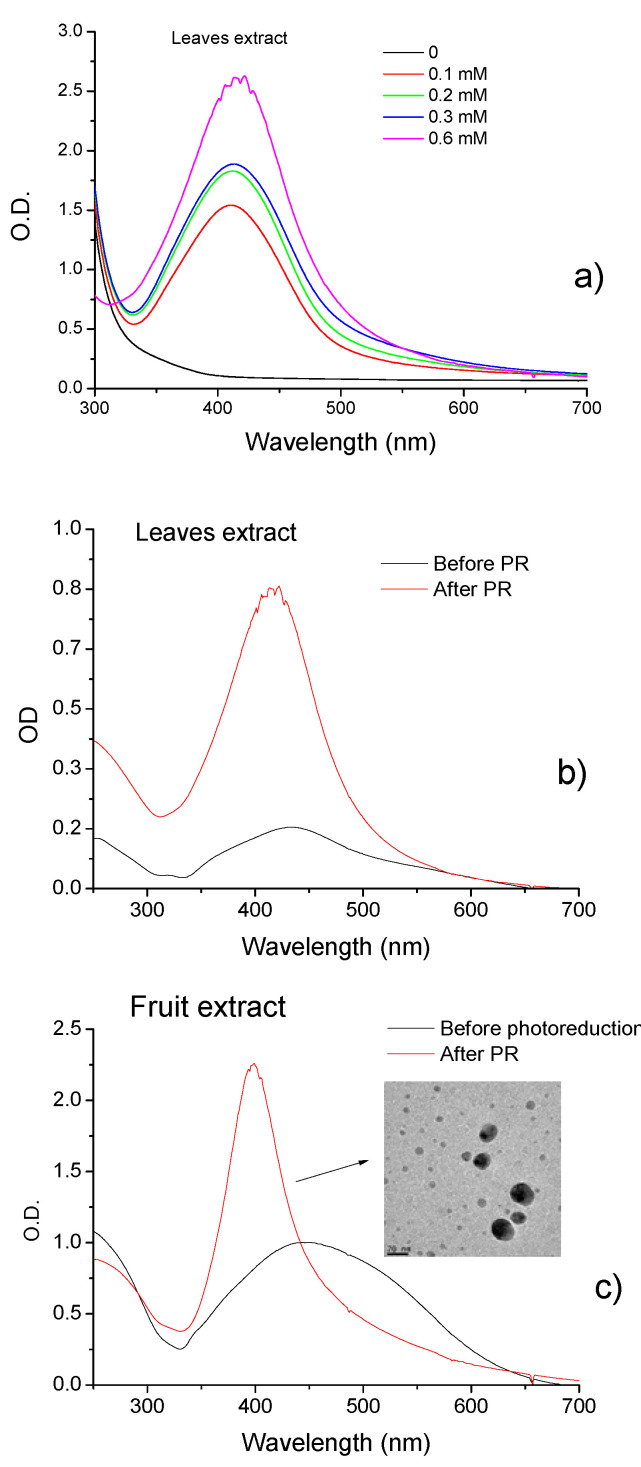
(**a**) PcAgNPs prepared with leaf extract SPR in the function of AgNO_3_ concentration. (**b**) UV-vis spectra of PcAgNPs before and after the photoreduction (PR) and pH adjustment for nanoparticles prepared with leaf extract, (**c**) with fruit extract and TEM image of PcAgNPs.

**Figure 4 molecules-27-06860-f004:**
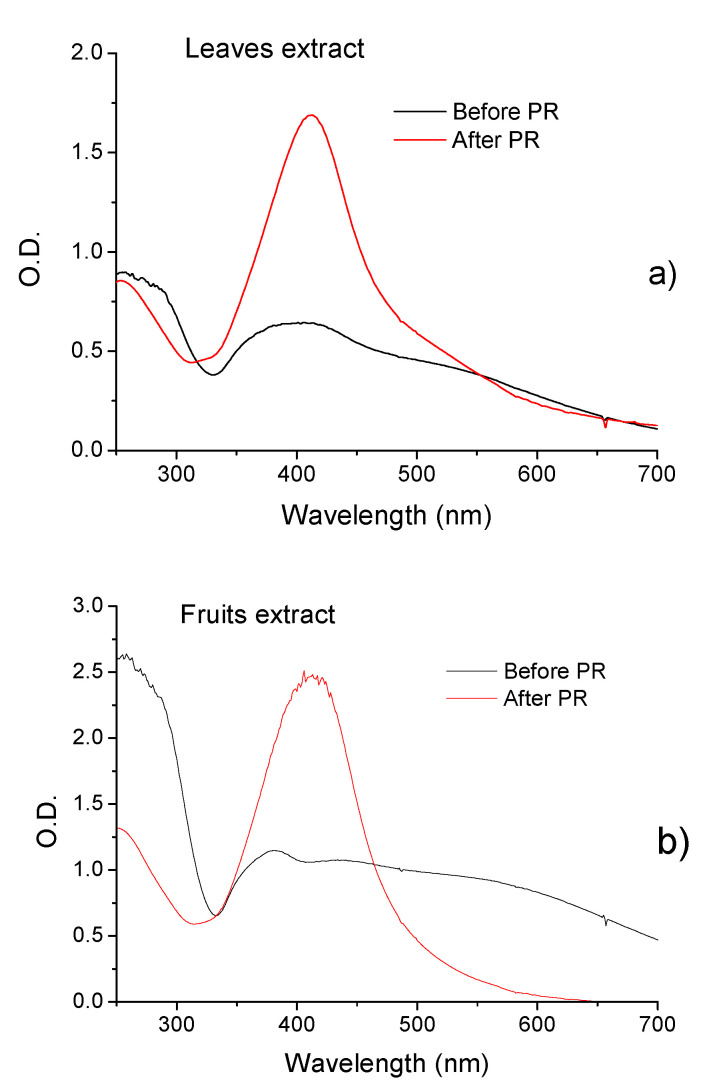
UV-vis spectra of PgAgNPs before and after the photoreduction and pH adjustment for nanoparticles prepared with (**a**) Leaf extract, (**b**) Fruit extract. TEM image of PcAgNPs.

**Figure 5 molecules-27-06860-f005:**
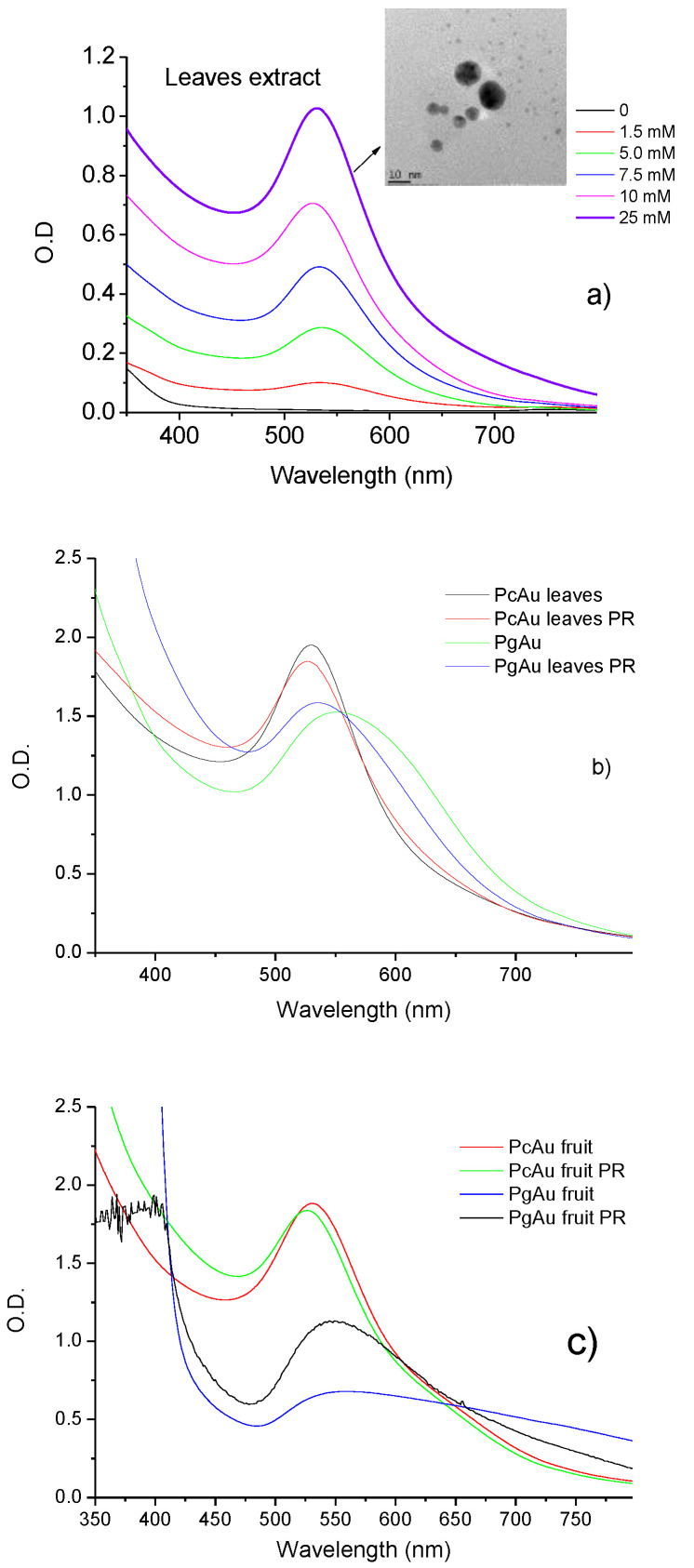
(**a**) PcAuNPs prepared with leaf extract SPR in the function of HAuCl_4_ concentration. TEM images PcAuNP. PcAuNPs and PgAuNPs prepared with photoreduction and pH 7.0. (**b**) leaf extract, and (**c**) fruit extract.

**Figure 6 molecules-27-06860-f006:**
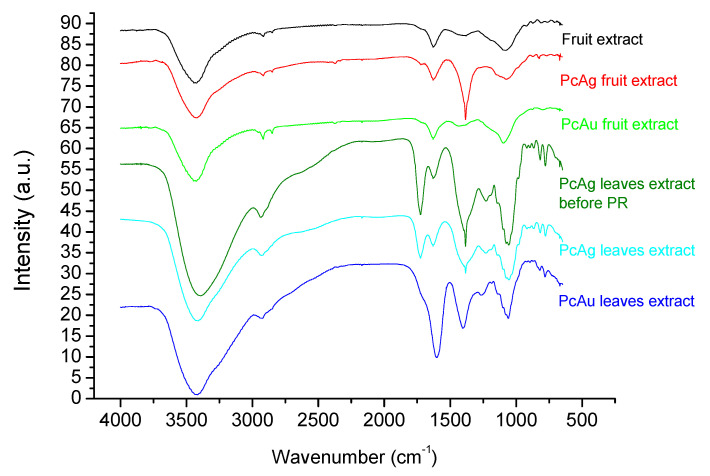
FTIR spectrum obtained from PcAgNP and PcAuNP prepared with leaves and fruit extract.

**Figure 7 molecules-27-06860-f007:**
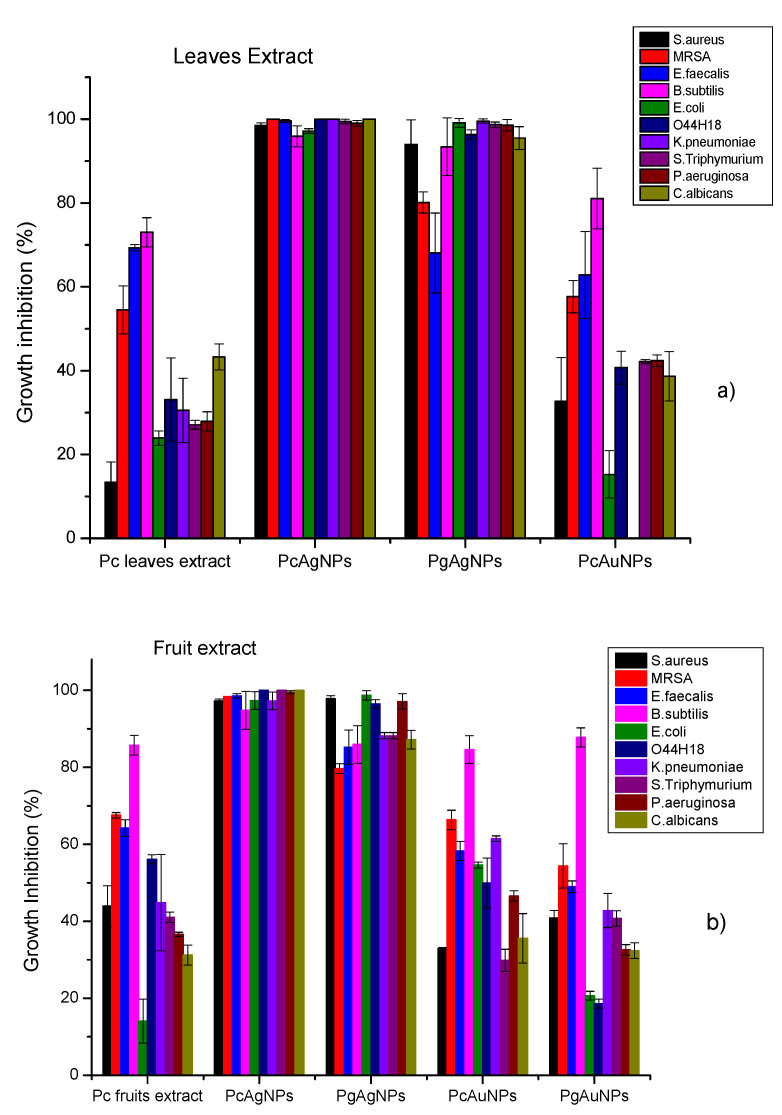
Antimicrobial activity of (**a**) leaves and (**b**) fruit extracts of *Plinia cauliflora* (“jabuticaba”) and their respective PcAg, PgAg, PcAu and PgAu nanoparticles against *Staphylococcus aureus* ATCC 25923, *Bacillus subtilis* ATCC 6633, methicillin-resistant *Staphylococcus aureus* (MRSA), *Enterococcus faecalis*, *Escherichia coli* ATCC 25922, *Escherichia coli* O44:H18 EAEC042, *Klebsiella pneumoniae* ATCC 700603, *Salmonella Thiphymurium* ATCC 14028, *Pseudomonas aeruginosa* ATCC 27853, and *Candida albicans* ATCC 10231 after 20 hours of incubation with PcAgNPs and PcAuNPs indicated in Table 1. The error bar denotes the standard error.

**Table 1 molecules-27-06860-t001:** Zeta potential, particle size, and PDI obtained for Pc and PgNPs. PR = photoreduction. All solutions with pH ~7.0.

Sample	Zeta Potential (mV)	Particle Size (±SD) (nm)	Polydispersivity Index (PDI)
PcAg leaf extract **before PR**	−10.8	66.84 ± 45.1	0.455
PcAg leaf extract, PR	−15.5	65.25 ± 31.22	0.252
PcAg fruit extract **before PR**	−17.1	84.79 ± 45.61	0.289
PcAg fruit extract, PR	−26.7	71.53 ± 36.56	0.261
PcAu leaf extract, PR	−17.1	84.79 ± 45.178	0.455
PcAu fruit extract, PR	−18.3	57.43 ± 40.78	0.499
PgAg leaf extract, PR	−20.1	80.71 ± 39.33	0.252
PgAg fruit extract, PR	−22.0	48.25 ± 30.00	0.387
PgAu fruit extract	−21.8	51.43 ± 36.20	0.469

## Data Availability

The data presented in this study are available on request from the corresponding author.

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
