# Peer review of "Antimicrobial Activity of Silver and Gold Nanoparticles Prepared by Photoreduction Process with Leaves and Fruit Extracts of Plinia cauliflora and Punica granatum"

_molecules, 2022, doi:10.3390/molecules27206860_

Round 1

Reviewer 1 Report

Authors reported Antimicrobial activity of silver and gold nanoparticles prepared by photoreduction process with leaves and fruits extracts of Plinia cauliflora and Punica granatum. Although Green Synthesis of Silver Nanoparticles is still an interesting, study, but the subject is to limited for a novelty. The journals receive a large number of review regarding on that subject. The authors must need to elaborate more to address the importance of this manuscript since the topic has been fairly active in research area

1.      Author must emphasize the novelty and orginality of the work in the abstract.

2.      What is the motivation behand this work, please give brief and clear explanation.

3.      Please clearly indicate advantages and disadvantages of plant extract based NP synthesis for comparison?

4.      Please provide potential mechanism for formation of NPs

5.      Please strictly revise the English language

6.      The articles about syntehesis of single or multi component noble metal NPs formed using various chemical and green synthsis methods (biomolecules including DNA, protein, enzyme and plant extracts) with their biological properties. Some of them should be considered for cite to improve the manuscript.

Adv. Mater. 25, 16, 2319-2325, 2013.
ACS Nano, 7, 10, 8972-8980, 2013.
and RSC Adv., 2020, 10, 22742-22757

Author Response

Reviewer 1:

Comments and Suggestions for Authors

Authors reported Antimicrobial activity of silver and gold nanoparticles prepared by photoreduction process with leaves and fruits extracts of Plinia cauliflora and Punica granatum. Although Green Synthesis of Silver Nanoparticles is still an interesting, study, but the subject is to limited for a novelty. The journals receive a large number of review regarding on that subject. The authors must need to elaborate more to address the importance of this manuscript since the topic has been fairly active in research area

  1. Author must emphasize the novelty and orginality of the work in the abstract.

This work describes two original insights: 1) the use of extracts of leaves and fruits from the Brazilian plant Plinia cauliflora and compared with a well know plant Punica granatum, and 2) the use of phytochemicals as stabilizing agents in the photoreduction process.

This text was included in the abstract.

  1. What is the motivation behand this work, please give brief and clear explanation.

The motivation behind this work was the obtention of silver and gold nanoparticles synthesized by the photoreduction process using the extract of plants and studying their potential as antimicrobial agents. This clean method allows the controlled generation of metallic nanoparticles.

  1. Please clearly indicate advantages and disadvantages of plant extract based NP synthesis for comparison?

The phytochemical agents present in plant extracts can reduce silver or gold to produce nanoparticles [48-52]. The advantages of plant extract-based nanoparticle synthesis are low-cost, safe, straightforward protocol, nanoparticles have high stability, generate non-toxic by-products, and allow large-scale synthesis. Some disadvantages can be listed: the process is not controlled, and particles are not monodispersed.

The photoreduction process employs light to promote the photochemical reaction and reduce metal ions to zero-valent metal, using the photochemically generated intermediates, such as excited molecules and radicals [53]. The advantages of this method are the absence of reducing agents, high spatial resolution, controllable generation of nanoparticles, and great versatility. This work employed the photoreduction process to synthesize silver and gold nanoparticles, and plant extracts were used as stabilizing agents.

This text was included in the “Introduction” section, lines 75-85.

  1. Please provide potential mechanism for formation of NPs

The mechanism for forming NPs by the photoreduction method is described in the “Discussion” section (lines 347-367) and illustrated in Figure 1.

 During the irradiation, mainly in the UV region, ketones present in the polyphenols structure are excited to the singlet excited state (Figure 1). The singlet excited state decays to the triplet excited state via the intersystem crossing. The excited triplets and subsequent hydrogen abstraction from the corresponding alcohol result in ketyl radicals which have been well-established as powerful reducing agents. These radicals reduce metal ions to generate metal nanoparticles [49].

In the case of Pc extract, there was an increased concentration of anthocyanins than for Pg extracts. The potential reaction between anthocyanin and Au3+ and AuNPs involves the formation of anthocyanins-Au3+ complexes, hydrogen (H+), and chloride (Cl-) ions[73]. With UV irradiation, charge transfer from anthocyanins to Au3+ as ligand-to-metal is induced, and Au+Cl2 species and quinone derivatives are produced. This process can explain the presence of a more intense SPR band for PcAuNPs compared with PgAuNPs.

  1. Please strictly revise the English language

English was revised.

  1. The articles about syntehesis of single or multi component noble metal NPs formed using various chemical and green synthsis methods (biomolecules including DNA, protein, enzyme and plant extracts) with their biological properties. Some of them should be considered for cite to improve the manuscript.

Adv. Mater. 25, 16, 2319-2325, 2013.

ACS Nano, 7, 10, 8972-8980, 2013.

and RSC Adv., 2020, 10, 22742-22757

References were included in the manuscript.

Author Response

The article studies the antimicrobial activity of silver and gold nanoparticles prepared by a photoreduction process in the presence of phytoextracts from leaves and fruits of Plinia cauliflora and Punica granatum.

The antibacterial activity was evaluated against Gram-negative and Gram-positive bacteria and yeast, showing better results for AgNPs than AuNPs.

The article could be very interesting, but it needs a thorough scientific revision because, in my opinion, it is quite superficial. In the following are some of my concerns:

  • Paragraph 2.2: I think a more detailed description of nanoparticle (NPs9 synthesis is necessary. What is the phytoextract pH? Is the irradiation carried out under stirring? What are the reagents used to adjust the final solution pH? Is the NPs solution neutralization performed immediately after the irradiation or not? How long does the synthesis last? How are the NPs treated to perform their characterization?

The required information were included in Section 2.2 “Preparation of nanoparticles”, lines 121-139“.

  • Paragraph 3.1: Did the authors evaluate the effect of irradiation on the phytoextract? This is very important for understanding their role in nanoparticle synthesis.

The irradiation on the phytoextract by 1 min does not promote a significant change in UV-Vis spectra. This information was included in the manuscript ( lines 196, 197).

3) NPs’ characterization: Why do the authors not show the systems regarding AuNPs without photoreduction as they made for AgNPs? I mean by comparing the system behavior with the two types of phytoextract as they made for Ag nanoparticle. In my opinion, the comparison could be interesting and, maybe, it could explain the main difference between the different systems.

Furthermore, did they perform a study of the AgNO3 concentration role?

Studies of AuNPs comparing the system behavior with the two types of phytoextract and a study of the AgNO3 concentration role were included, as proposed by the reviewer (Figures 3 and 5) and correspondent text.

  • Table 1: Table 1 is quite confusing. To what sample do the authors refer using “without photoreduction”? The paper deals with two different metallic nanoparticles prepared with and without photoreduction. Although the absorption spectra can be considered quite clear, the data of Z potential are not. Please clarify.

Table 1 and text were modified.

5) FTIR experiments: This paragraph is very superficial. I don’t understand why the authors added these experiments. Did they want to understand the functional groups involved in the NPs formation? Did they want to underline some different characteristics between the two systems? Again, I don’t understand. What do they want to demonstrate?

Moreover, it is true that other authors previously discussed the FTIR features of the phytoextract, but in my opinion, a more detailed discussion should be present also in this case. In fact, the authors have chosen two different plants characterized by different components that act differently in the synthesis and probably confer different properties to the metallic nanoparticles. This aspect should be discussed eventually also by comparing the different FTIR spectra. Another aspect I don’t understand is the statement regarding the signal at 1384 cm-1. The authors affirm that this peak is absent in the AuNPs FTIR spectrum, but I don’t see it this way. In the region around 1300-1400 cm-1, for AuNPs (blue line in figure 6), there is a strong, intense signal that may hide the ess intense 1384 cm-1 peak.

Therefore, I think that this paragraph needs a thorough scientific revision considering to add the FTIR spectra of the phytoextract.

Figure 6 presents the Fourier transform infrared spectroscopy (FTIR) obtained for the phytoextract [72] [73] and PcAgNPs and PcAuNPs. The spectra present strong and broad peaks around 3000 to 3600 cm−1 correspondents to the -NH2 (amide I) and/or -OH of phenolic compounds. The sharp doublet peaks at 2920 cm−1 and 2845 cm−1 (fruit extract) correspond to the symmetric and asymmetric vibrational mode of -CH stretching. The spectral region between 1600 and 1700 cm−1 is related to aromatic C=O stretching vibration of carbonyl. The absorption peak around 1733 cm-1 due to the C=O stretching vibration absorption of flavonoids and amides is observed in PcAg prepared leaves extract. After the photoreduction process, a reduction in this band intensity, indicating that C=O bond was oxidized[74]. The sharp peak at 1384 cm−1 is due to C-H stretching vibrations of aromatic and aliphatic amines. The peak around 1060 cm-1 indicates C-O stretching vibrations correspond to the presence of alcohols, carboxylic acids, ethers, and esters. So, the IR spectra of prepared NPs thus confirmed that the carbonyl group of polyphenols can bind metal, indicating that the biological molecules could perform both functions of formation and stabilization of silver nanoparticles in the aqueous medium. 

This text was included in the manuscript (lines 260-274).

6) Paragraph 3.3 (lines 249-253): There is confusion in the symbols used to distinguish the different samples. Please also pay attention to the symbols that identify the various samples, both to the figures’ legends and the figure caption.

Section 3.3 was revised.

7) Paragraph 3.3: Where is the Pg leaves extract? Did the authors use it or not? Why did the authors not show the results?

Unfortunately, we haven’t this result since they were already investigated by other authors.

8) Discussion: The whole paragraph regarding the results’ discussion needs a scientific revision. It is confusing, and there are many repetitions. It lacks its role because many issues remain undiscussed and so not clear.

For example, how does the photochemical process claimed by the authors modify the NPs’ synthesis? What is the role of the solution pH in NP formation? Is the solution pH important for the photochemical process? Again, what is the photostability of the phytoextract under irradiation? Are the authors sure that the final oxidation state of the metallic nanoparticles is Au(0) and Ag(0)?

Probably some XPS analysis could be useful to clarify these aspects.

The “discussion” section was changed to answer the referee’s questions.

We appreciate the remarks.

Round 2

Reviewer 1 Report

responses are satisfactory

Reviewer 2 Report

The manuscript in this new form is publishable.